# Regulations of Citrus Pectin Oligosaccharide on Cholesterol Metabolism: Insights from Integrative Analysis of Gut Microbiota and Metabolites

**DOI:** 10.3390/nu16132002

**Published:** 2024-06-24

**Authors:** Haijuan Hu, Peipei Zhang, Fengxia Liu, Siyi Pan

**Affiliations:** 1College of Food Science and Technology, Huazhong Agricultural University, Wuhan 430070, China; huhaijuan@163.com (H.H.); zhangpeipei1217@webmail.hzau.edu.cn (P.Z.); liufxia@mail.hzau.edu.cn (F.L.); 2Key Laboratory of Environment Correlative Dietology, Ministry of Education, Huazhong Agricultural University, Wuhan 430070, China

**Keywords:** citrus pectin oligosaccharides, cholesterol metabolism, gut microbiota and metabolites

## Abstract

(1) Background: Recently, academic studies are demonstrating that the cholesterol-lowering effects of pectin oligosaccharides (POSs) are correlated to intestinal flora. However, the mechanisms of POS on cholesterol metabolisms are limited, and the observations of intestinal flora are lacking integrative analyses. (2) Aim and methods: To reveal the regulatory mechanisms of POS on cholesterol metabolism via an integrative analysis of the gut microbiota, the changes in gut microbiota structure and metabolite composition after POS addition were investigated using Illumina MiSeq sequencing and non-targeted metabolomics through in vitro gut microbiota fermentation. (3) Results: The composition of fecal gut flora was adjusted positively by POS. POS increased the abundances of the cholesterol-related bacterial groups *Bacteroidetes*, *Bifidobacterium* and *Lactobacillus*, while it decreased conditional pathogenic *Escherichia coli* and *Enterococcus*, showing good prebiotic activities. POS changed the composition of gut microbiota fermentation metabolites (P24), causing significant changes in 221 species of fermentation metabolites in a non-targeted metabolomics analysis and promoting the production of short-chain fatty acids. The abundances of four types of cholesterol metabolism-related metabolites (adenosine monophosphate, cyclic adenosine monophosphate, guanosine and butyrate) were significantly higher in the P24 group than those in the control group without POS addition. (4) Conclusion: The abovementioned results may explain the hypocholesterolemic effects of POS and promotion effects on cholesterol efflux of P24. These findings indicated that the potential regulatory mechanisms of citrus POS on cholesterol metabolism are modulated by cholesterol-related gut microbiota and specific metabolites.

## 1. Introduction

As an indispensable part of the human body, gastrointestinal microflora is of importance to human health, nutrient metabolism and disease development [1,2,3,4]. In general, the major gut microbial phyla include *Firmicutes*, *Bacteroidetes*, *Actinobacteria*, *Proteobacteria*, *Fusobacteria* and *Verrucomicrobia*, with *Firmicutes* and *Bacteroidetes* being the two most abundant phyla [5,6,7]. Strong correlations were observed between changes in cholesterol levels and microbiota composition [8,9,10]. The abundance of fecal *Bifidobacterium* and *Bacteroides* in hypercholesterolemia patients is negatively correlated with changes in blood cholesterol levels [8]. Sudun et al. found that probiotics alleviated high-fat diet-induced hypercholesterolemia by regulating the gut microbiota [9]. In addition, mannan oligosaccharides have been suggested to regulate cholesterol metabolism via gut microbiota [10]. Thus, adjusting the gut microbiota composition has been regarded as a promising strategy to modulate cholesterol metabolism.

The essential role of cholesterol metabolism-related gut flora and their microbial metabolites is receiving attention [11,12,13,14,15,16]. For gut microflora, *Bacteroidetes*, as a symbiotic gut bacterium, has been proven to possess cholesterol-lowering properties [11]. Moreover, the main probiotics, lactobacilli and bifidobacteria, are reported to be effective in reducing cholesterol levels [12,13]. Various metabolites could be produced in gut microbiota metabolism [14,15,16]. A large body of evidence has shown that intestinal bacteria metabolites regulate the host metabolism positively [17,18]. Bile acids synthesis is reported to be improved by regulating the transcriptional activity of FXR [19]. As the three main short-chain fatty acids (SCFAs) of gut microbiota metabolites, acetic acid, propionic acid and butyric acid were found to reduce plasma total cholesterol (TC) levels in hamsters by promoting cholesterol decomposition and efflux [20]. The adenosine triphosphate (ATP)-binding cassette transporter A1 (ABCA1) is one of the key mediators of macrophage cholesterol efflux [21]. Cholesterol metabolism could be promoted by upregulating the expression of the ABCA1 gene to enhance cholesterol efflux [21]. Butyric acid has been reported to promote cholesterol efflux in HF-diet ApoE knockout mice by upregulating ABCA1 gene expression in macrophages [22].

The intestinal microbiota composition was reported to be selectively stimulated by prebiotics to confer health effects [23]. As potential prebiotics, pectin and chitosan oligosaccharides have been proven to promote cholesterol metabolism by modifying intestinal flora compositions and SCFA profiles [24,25]. According to our previous study, citrus pectin oligosaccharide (POS), previously referred to as POS_H1_, prepared from a novel chemically controllable degradation method, is a potential prebiotic [26]. Our findings indicated that the hypocholesterolemic effects of POS_H1_ were related to specific gut bacterial groups and their metabolites [27]. Furthermore, microbial metabolites of POS_H1_ (previously referred to as P24) have been demonstrated to promote cholesterol efflux and inhibit cholesterol uptake and synthesis, as reported in our previous research [28]. However, the mechanisms of POS_H1_ on cholesterol metabolisms via an integrative analysis of gut microbiota structure and P24 composition remain unknown. Therefore, the aims of this study were to (i) characterize the intestinal flora structure after POS_H1_ intervention, (ii) analyze the composition of P24 and (ii) reveal the possible regulatory mechanism of POS_H1_ on cholesterol metabolism via gut microflora and their metabolites.

## 2. Materials and Methods

### 2.1. Chemicals and Reagents

POS_H1_ was prepared from a novel chemical controllable degradation method, according to our previous study [26]. TransStart FastPfu Fly PCR SuperMix was purchased from Transgen Biotech (Beijing, China). E.Z.N.A.^®^ soil DNA kit was from Omega (Guangzhou, China), and AxyPrep DNA Gel Recovery Kit was from Axygen Biosciences (Union City, CA, USA). Acetic acid, propionic acid, butyric acid, isobutyric acid, valeric acid and isovaleric acid were of chromatographic grade from Sigma-Aldrich (St. Louis, MA, USA). Chromatographic-grade methanol, acetonitrile, formic acid, propanol and ultrapure water were all purchased from Thermo Fisher Scientific (Waltham, MA, USA), except for L-2-chlorophenylalanine, which was from Aladdin (Shanghai, China).

### 2.2. In Vitro Fermentation

In vitro fermentation was performed according to our previous report [28]. Detailed methods are included in the in the Appendix A. Supernatants were defined as P24 (background medium without POS_H1_: P24_1, P24_2, P24_3, P24_4, P24_5 and P24_6) or N24 (background medium with POS_H1_ substrate: N24_1, N24_2, N24_3, N24_4, N24_5 and N24_6), respectively. The sediment of bacterial sludge obtained from the control group was denoted as Group N, which consisted of 6 samples (N1, N2, N3, N4, N5 and N6). Sediments of bacterial sludge obtained from the POS_H1_ group were denoted as Group P, which consisted of 6 samples (P1, P2, P3, P4, P5 and P6).

### 2.3. DNA Extraction, PCR Amplification and Illumina MiSeq Sequencing

E.Z.N.A.^®^ soil DNA Kit was used to extract microbial community genomic DNA from bacterial sludge sediments. Details regarding the methods employed for the DNA extraction, PCR amplification and Illumina MiSeq sequencing are provided in the Appendix A. Each assay was repeated three times.

### 2.4. Processing of Sequencing Data

The raw 16S rRNA gene sequencing reads were demultiplexed, quality-filtered by fastp version 0.20.0 [29] and merged by FLASH version 1.2.7 with the following criteria [30]. Detailed analyses of the processes are shown in the Appendix A [31,32,33].

### 2.5. Metabolite Extraction and UPLC-MS/MS Analysis

Metabolites were extracted and characterized through UPLC-TOF/MS analysis. Additional details are included in the Appendix A. Each assay was replicated a minimum of three times.

### 2.6. Data Preprocessing and Annotation

After the UPLC-TOF/MS analyses, Progenesis QI 2.3 (Waters, MA, USA) was used for peak detection and alignment of the raw data. Accurate mass, MS/MS fragment spectra and isotope ratio differences, along with searching in reliable biochemical databases such as the Human Metabolome Database (HMDB) and the Metlin database, were used to identify the mass spectra of these metabolic features. Details are given in the Appendix A.

### 2.7. Multivariate Statistical Analysis

Majorbio Cloud Platform was employed to perform a multivariate statistical analysis. Principle component analysis (PCA) using an unsupervised method was applied to obtain an overview of the metabolic data; general clustering, trends or outliers were visualized. Partial least squares discriminate analysis (PLS-DA) was used for a statistical analysis to determine the global metabolic changes between comparable groups. Details are provided in the Appendix A.

### 2.8. Differential Metabolites Analysis

Statistically significant results among groups were selected according to the VIP value (>1) and *p*-value (<0.05). Metabolic enrichment and a pathway analysis based on database search were utilized to summarize and map differential metabolites among two groups into their biochemical pathways. Details are given in the Appendix A.

### 2.9. Determination of the Concentration of SCFAs in Intestinal Flora Fermentation Products

The detection of SCFAs was analyzed using gas chromatography−mass spectrometry. Details are provided in the Appendix A. Each assay was performed in triplicate.

### 2.10. Statistical Analysis

The Wilcoxon rank-sum test was used to analyze the significant difference between the α-diversity index and the intestinal microbial species group, and the ANOSIM test was used to analyze the significant difference between the β-diversity index groups. The difference between the supernatant fermentation products was analyzed by Student’s *t*-test. The Kyoto Encyclopedia of Genes and Genomes (KEGG) pathway enrichment analysis was performed on the differential fermentation products; Fisher’s exact test was used for the enrichment analysis, and Benjamini and Hochberg were selected to verify the *p*-value; and the false positive of the enrichment results was controlled; and Benjamini was used by default, and the Hochberg method corrected the *p*-values. The corrected *p*-value takes 0.05 as the threshold, and KEGG pathways satisfying this condition are defined as KEGG pathways that are significantly enriched in the metabolic set. All the experimental data of SCFAs were repeated three times independently, and the results were expressed as mean ± standard error of the mean (SEM). Excel software 2021 was used to sort out the experimental data, and GraphPad Prism 7.0 software was used for the analysis. The *t*-test was used for a comparison between the two groups, and *p* < 0.05 indicated that the difference was statistically significant.

One-way ANOVA analysis was used to compare the differences between the means of each group, and an LSD post hoc test was used to compare the differences between the two groups. The difference was statistically significant when *p* < 0.05. The data analysis software was SPSS v. 21.0 software (IBM, NY, USA).

## 3. Results

### 3.1. Changes in the Dilution Curve

A total of 589,656 optimized sequences (with an average sequencing depth of 49,138) were obtained by the high-throughput sequencing of 12 samples in Group P and Group N. Operational taxonomic unit (OTU) clustering was performed on the optimized sequence, with a threshold of 97% similarity; a total of 401 OTUs were obtained, and the average coverage of these OTUs for all sequences in the sample was 99.9%. At the current sequencing depth, the Shannon exponential dilution curves of all samples have reached a plateau, as shown in Figure 1. As shown in the dilution curves of intestinal microorganisms in Groups P and N, the dilution curve tended to be flat as the number of sequences increased and the number of species increased to a constant level. The abovementioned results indicated that the sequencing results were reliable and that the vast majority of species in the sample have been covered. The intestinal microbial dilution curves of the P group and the N group demonstrated that the intestinal microbial diversity of the P group was higher than that of the N group, indicating that the addition of POS_H1_ to the control group significantly increased the intestinal microbial diversity.

### 3.2. The Effects of POS_H1_ on the α-Diversity of Intestinal Microorganisms

The effect of POS_H1_ on the α-diversity of intestinal microorganisms is shown in Figure 2. POS_H1_ significantly increased the Shannon index (*p* < 0.01) when compared to the control group, and the Shannon index was positively correlated with species diversity. POS_H1_ significantly decreased the Simpson index (*p* < 0.01) when compared to the control group, and the Simpson index value was negatively correlated with community diversity. POS_H1_ significantly increased the Chao index (*p* < 0.01) compared to the control group, and the Chao index was positively correlated with community species richness. POS_H1_ significantly increased the Ace index (*p* < 0.01) compared to the control group, and the Ace index was positively correlated with the species richness of the community. Therefore, the gut microbes of the POS_H1_ group exhibited higher alpha diversity.

### 3.3. The Effects of POS_H1_ on Intestinal Microbial β Diversity

Based on the Bray–Curtis distance, a cluster analysis was performed on all samples, as shown in Figure 3A. The results of the sample-level cluster analysis showed that the composition of bacterial communities in Group P and the control group, Group N, was significantly different. To further investigate the effects of adding POS_H1_ to the medium on the overall structure of intestinal microorganisms, the principal component analysis (PCA) was used as an evaluation index for the β-diversity of intestinal microorganisms, and the results are shown in Figure 3B. The PCA analysis indicated that the distribution of samples between the P group and the N group was far away, and the similarity between groups with clear boundaries was low, indicating that a significant difference in the intestinal flora composition between the two groups was observed and that the composition of intestinal bacteria in the samples of the two groups was significantly different (*p* = 0.003).

### 3.4. The Effects of POS_H1_ on the Abundance of Gut Microbial Phylum, Genus and Species

The effects of POS_H1_ on gut microbes at the phylum level and significant differences between groups are shown in Figure 4A and 4B, respectively. The dominant phyla of intestinal microorganisms in Group N and Group P were *Proteobacteria*, *Firmicutes* and *Bacteroidetes*, but their abundance ratios were different. The relative abundances of *Firmicutes*, *Bacteroidetes* and *Actinobacteria* in Group P were 51.20%, 20.56% and 9.58%, respectively, which were significantly higher than those in Group N (*Firmicutes*, 17.14%; *Bacteroidetes*, 9.95%; and *Actinobacteria*, 0.45%) (both *p* = 0.005). Among them, the promotional effect of POS_H1_ on the abundance of *Firmicutes* and *Bacteroidetes* is consistent with the findings of our previous report [27]. The abundances of *Proteobacteria* and *Fusobacteria* in Group P were 15.97% and 2.40%, respectively, which were significantly lower in Group N (*Proteobacteria*, 65.82%, *p* < 0.01; and *Fusobacteria*, 6.487%, *p* < 0.05).

The effects of POS_H1_ on gut microbes at the genus level and significant differences between groups at the genus level are shown in Figure 5A and 5B, respectively. *Escherichia coli* is the main member of the *Proteobacteria* phylum, and the abundance of *Escherichia–Shigella* in Group P was 5.28%, which was 10% of the abundance in Group N (51.85%) (*p* = 0.005). The genus *Megamonas* was a member of *Firmicutes*, and its abundance in Group P (20.53%) was significantly higher than that in Group N (2.57%) (*p* = 0.005). *Prevotella* is one of the main genera of *Bacteroidetes*, and its abundance in Group P was 5.75 times higher than that in Group N (2.39%); its abundance was 13.71% (*p* = 0.005). *Bifidobacterium* is the main genus in *Actinomycetes*, and its abundance in Group P (0.45%) was significantly higher than that in Group N (0.03%), which was 13 times that of Group N (*p* = 0.005). This result is consistent with the growth-promoting effect of POS_H1_ on bifidobacteria noted in our previous reports [20,21]. The abundance of *Enterococcus* in Group P (0.23%) was significantly lower than that in Group N (1.8%) (*p* = 0.005), which is consistent with the effect of POS_H1_ on fecal contents in vivo in our previous observations [27]. The trend of the change in the number of *Enterococcus* was consistent. In addition, the diversity results showed that small amounts of *Lactobacillus* (0.003%) and *Akkermansia* (0.006%) were detected in the P group, while the presence of these two bacteria was not detected in the N group. Among them, the *Lactobacillus* in Group P was significantly higher than that in Group N (*p* < 0.05), and this result was in agreement with the growth-promoting effect of POS_H1_ on lactic acid bacteria found in our previous research [27].

The effects of POS_H1_ on the species level of intestinal microorganisms and significant differences between groups at the species level of microorganisms are shown in Figure 6A and 6B, respectively. The abundance of *Escherichia coli* in Group P is 5.27%, while 10% of its abundance in Group N (51.72%) (*p* = 0.005), which is consistent with the growth inhibitory effect of POS_H1_ on *Escherichia coli* in our previous report [27]. The changes at the species level of *Megamonas* and *Prevotella* in Groups P and N were consistent with the above-mentioned alterations at the genus level. The relative abundance of *Enterococcus faecium* in Group P was 0.18%, which was significantly lower than that in Group N (1.37%). The abundances of *Bifidobacterium pseudocatenulatum* and *Bifidobacterium longum* of the *Bifidobacterium* genus in Group P was significantly higher than that in Group N, as they were 17.8 and 5.5 times that of Group N (*p* < 0.05). The growth-promoting effect of POS_H1_ on bifidobacteria was in agreement with our previous reports [26,27].

### 3.5. The Effect of POS_H1_ on the Composition of Fermentation Products of Intestinal Flora

The metabolomics results demonstrated that a total of 6217 ions were identified in the positive ion mode, and 5312 in the negative ion mode. During the analysis process, three QC tests were performed on the samples. After removing low-mass ions (relative standard deviation > 30%), 5217 and 4721 ions were identified in the positive and negative ion modes, respectively. As shown in Figure 7, the results of the partial least squares discriminant analysis (PLS-DA) showed that the R2 and Q2 of the positive and negative ion score maps were both close to 1, indicating that the model was stable and reliable and had good predictive ability. Under this model, the supernatant fermentation products of the P24 and N24 groups were well separated.

The principal component analysis (PCA) results of the supernatant samples of the P24 and N24 groups are shown in Figure 8. The PCA analysis indicated that the non-targeted metabolomics data of the P24 and N24 groups had a clear boundary; the sample distribution between the groups was far away and the similarity was low. Those results indicated that the composition of the fermentation products of the two groups was significantly different (*p* < 0.05), showing that the addition of POS_H1_ to the medium caused metabolic changes in the overall gut microbiota.

The mass spectrometric identification results of supernatant metabolome showed that 593 kinds of fermentation products were identified in the supernatants of the P24 and N24 groups. To further investigate and explain the influence of POS_H1_ on the composition of supernatant fermentation products, the difference of fermentation products in the supernatant P24 and N24 of the two groups was analyzed via screening with a *p* < 0.05 and VIP > 1, and a total of 221 species were obtained in the two groups. Compared with the control group, Group N24, the abundance of 121 fermentation products was elevated and 100 fermentation products decreased in the P24 group with POS_H1_ fermentation.

### 3.6. Analysis of Differential Fermentation Products

The mass spectrum results of differential fermentation products were matched with the HMDB database, and the number of differential fermentation products was classified. The three major categories of substances are acids and their derivatives and organic heterocyclic compounds (Figure 9).

The mass spectrum information of differential fermentation products was matched with the KEGG database, and the differential fermentation products were classified according to the involved pathways. As shown in Figure 10, seven categories can be divided in KEGG metabolic pathways, namely metabolism, genetic information processing, environmental information processing, cellular processes, biological systems, human diseases and drug development.

Taking the collection of all fermentation products of this species as the enrichment background, a KEGG pathway enrichment analysis was performed on the differential fermentation products. Fisher’s exact test was used for the enrichment analysis, and with a *p* < 0.05 as the threshold, 68 metabolic pathways with significant enrichment of differential fermentation products could be screened, as shown in Figure 11. The above 68 significantly enriched pathways consisted of 38 differential fermentation products, as shown in Table 1. Among them, three kinds of cholesterol metabolism-related metabolites were detected: adenosine monophosphate, cyclic adenosine monophosphate (cAMP) and guanosine nucleoside. Their abundance in the P24 group was significantly higher than that in the N24 group.

### 3.7. The Effects of POS_H1_ on SCFAs in the Fermentation Products of Intestinal Flora

The effects of POS_H1_ on SCFAs in the fermentation products of intestinal flora are shown in Figure 12. The composition of SCFAs was similar in the N24 and P24 groups. Acetic acid was the highest content of all SCFAs, followed by butyric acid (the sum of butyric acid and isobutyric acid) and propionic acid, and the lowest content was valeric acid (the sum of valeric acid plus isovaleric acid). The P24 group significantly increased the levels of the detected six SCFAs in the fermentation products of the intestinal flora compared to the control group, Group N24, which was similar with the phenomenon observed in previous study [27].

## 4. Discussion

This study investigated the cholesterol-lowering mechanisms of POS_H1_ via an integrative analysis of the gut microbiota. The findings indicated that POS promoted the growth of the cholesterol-related bacterial groups *Bacteroidetes*, *Bifidobacterium* and *Lactobacillus* and increased four types of cholesterol metabolism-related metabolites (adenosine monophosphate, cyclic adenosine monophosphate, guanosine and butyrate) concentrations in non-targeted metabolomics and SCFA analyses. Our results demonstrated that the potential regulatory mechanisms of citrus POS on cholesterol metabolism are modulated by cholesterol-related gut microbiota and specific metabolites.

The microecological balance of the intestinal microbial composition played an important role in maintaining body homeostasis [31]. A β-diversity analysis was used to analyze the similarity or difference relationship of microbial community structures in different samples. The results of the β-diversity showed that the intervention of POS_H1_ caused changes in the intestinal microorganisms. Alpha diversity was used to analyze the diversity and species richness of different samples [32,33]. The results of α-diversity in this study showed that POS_H1_ significantly increased the Shannon, Chao and Ace indices and significantly decreased the Simpson indices compared to the control group. In conclusion, the addition of POS_H1_ significantly increased the diversity and richness of intestinal microorganisms.

Our study demonstrated that POS_H1_ significantly altered the composition of fecal gut microbiota, inhibited the growth of potential pathogenic bacteria, and promoted the growth of beneficial bacteria. At the phylum level, *Proteobacteria* showed the highest abundance in the control group in the dominant phylum, followed by *Firmicutes* and *Bacteroidetes*, while *Firmicutes* reflected the highest abundance in the POS_H1_ group, followed by *Bacteroidetes* and *Actinobacteria*; and *Proteobacteria* with *Fusobacteria* were relatively less abundant bacteria. *Proteobacteria* was an opportunistic pathogen, including *Escherichia coli*, *Salmonella* and *Campylobacter*. *Proteobacteria* has been reported as a potential factor resulting in intestinal diseases, and the increased *Proteobacteria* abundance increased the prevalence of intestinal diseases risk [34,35]. In the current study, POS_H1_ significantly reduced the abundance of *Proteobacteria*, which was consistent with the report in the literature that pectin reduced the abundance of *Proteobacteria* in pig intestinal flora [36], indicating that POS_H1_ has potentially regulated intestinal flora and reduced the risk of intestinal diseases. The abundance of *Firmicutes* was significantly increased in the POS_H1_ group and became the dominant phylum with the highest abundance. The promotional effect of POS_H1_ on the abundance of *Firmicutes* was consistent with the experimental phenomenon observed in our previous study [27]. Similar results have been reported in related studies. The lemon pectin treatment group significantly promoted the abundance of *Firmicutes*, which became the most abundant phylum in the human intestinal flora [37]. Our results of the increased growth of *Bacteroidetes* and *Actinobacteria* stimulated by POS_H1_ are in accordance with previous studies [38,39]. Moreover, the increased abundance of cholesterol metabolism-related microbiota *Bacteroidetes* after POS_H1_ supplementation is in agreement with observations made in our previous study [27]. The structural analysis performed in our previous study showed that POS_H1_ was a low-esterification pectin oligosaccharide [28]. Larsen et al. reported that the lower the degree of pectin esterification, the easier it is to be decomposed and utilized by *Bacteroidetes*, and the better it is to promote the growth of *Bacteroides* [40], which may be the explanation for the increased abundance of the *Bacteroidetes* phylum by POS_H1_.

At the genus and species levels, POS_H1_ significantly inhibited the growth of *Escherichia–Shigella* and *Escherichia coli*, and the abundance was only 10% of that of the control group. The *Escherichia–Shigella* genus contains many potential pathogenic bacteria, including *Escherichia coli*, and a reduced abundance of this flora has been considered to be beneficial to the health of the intestinal flora [41,42,43]. The reduction in the abundance of *Escherichia coli* may be related to the protective effect of *Bifidobacteria* on the intestinal tract. *Bifidobacteria* has been proven to inhibit the growth of *Escherichia coli* by producing acetic acid [44]. *Enterococcus* is the main pathogen that causes body infection, and the inhibited growth of *Enterococcus* was reported to protect the intestinal flora and regulate the intestinal microecology [45,46]. POS_H1_ significantly reduced the relative abundance of *Enterococcus* in the intestinal flora and inhibited its growth, which was consistent with the inhibition trend of POS_H1_ on *Enterococcus* in mice fecal contents in our previous report [27]. *Prevotella* is one of the major genera of *Bacteroidetes* with anti-inflammatory effects [47]. POS_H1_ significantly increased the abundance of *Prevotella* compared with the control group. *Bacteroides* and *Prevotella* are the main flora that decompose and utilize pectin in the intestine, and the intake of pectin and dietary fiber foods significantly increases the abundance of *Prevotella* [36,40,48,49]. *Bifidobacterium* and *Lactobacillus* are two types of probiotics recognized in the intestinal flora that exert beneficial effects by modulating cholesterol metabolism [50,51], POS_H1_ significantly increased the relative ratio of the two in the intestinal flora abundance, especially the relative abundance of *Bifidobacterium pseudocatenulatum* and *Bifidobacterium longum* in the *Bifidobacterium* genus, similar to the prebiotic effects of pectin and pectin oligosaccharides reported in the literature [52,53,54]. In addition, studies have shown that the cross-feeding mechanism between *Bacteroidetes* and *Bifidobacterium* genera promotes the growth and reproduction of Bifidobacterium [55]. These findings are consistent with the growth-promoting effect of POS_H1_ on cholesterol metabolism-related intestinal flora, lactic acid bacteria and bifidobacteriaobserved in the in vivo research in our previous study [27]. It showed that POS_H1_ promoted the growth of beneficial bacteria in the intestinal flora and exerted potential probiotic activity.

Intestinal flora is a highly complex micro-ecological system. In addition to the composition of the flora itself affecting the health of the host, its fermentation products are also an important way for the intestinal flora to regulate the host metabolism [18,56]. The fermentation products produced by the intestinal flora after the fermentation of POS_H1_ caused a total of 221 significant changes in the fermentation products, and the metabolic pathways of the differential fermentation products were mainly concentrated in lipids and lipid-like substances, organic acids and their derivatives, and organic heterocyclic compounds.

Small molecules in fermentation products produced by intestinal flora, such as BAs, SCFAs, amino acids, and other derivatives, have been reported to be involved in regulating the body’s physiological metabolism and maintaining host health [57]. The results of KEGG pathway enrichment showed that 68 significantly enriched pathways were composed of 38 differential fermentation products. Among them, there were three kinds of fermentation products related to cholesterol metabolism, namely adenosine monophosphate (adenosine monophosphate), cAMP (cyclic adenosine monophosphate) and guanosine (guanosine nucleoside), the abundance of which was significantly higher in the P24 group than in the N24 group.

Adenosine monophosphate-activated protein kinase (AMPK) is involved in the regulation of multiple cellular biological processes, including cholesterol metabolism. When the ratio of adenosine monophosphate to adenosine triphosphate increased, AMPK could be activated by phosphorylating the T172 binding site [14]. Studies have shown that AMPK prevented the development of atherosclerosis-related cardiovascular disease by upregulating the expression of ABCA1 and ABCG1 in macrophages and promoting HDL-regulated cholesterol efflux in macrophages [58]. Therefore, compared with N24, the increased adenosine monophosphate content in P24 might explain the reason for significantly promoting the effects of P24 on the cholesterol efflux of macrophages in our previous reported study [28]. After cyclic adenosine monophosphate treatment of macrophages, the mRNA expression of ABCA1 in the cells increased to 4.1 times, which significantly promoted the cholesterol efflux [59]. Therefore, the cholesterol efflux-promoting effects of P24 reported in our previous study might be associated with the marked rise of cyclic AMP in the P24 composition [28]. Guanosine has been reported to be able to increase cholesterol efflux in astrocytes and C6 rat glioma cells without exogenous receptors [60]. In addition, significantly elevated guanine in P24 components might also be another reason why P24 significantly promoted the cholesterol efflux of macrophage at a normal state in our previous study [28].

SCFAs are the main end products of indigestible sugars fermented by intestinal microorganisms, mainly including acetic acid, propionic acid, isobutyric acid, butyric acid, isovaleric acid and valeric acid, among which acetic acid, propionic acid and butyric acid account for the sum of SCFAs by more than 95% [61]. In the current study, acetic acid, propionic acid and butyric acid were the most important SCFAs in the P24 and N24 groups, and valeric acid and isovaleric acid accounted for only 4.9–7.2%. Studies have shown that pectin and its hydrothermal hydrolyzed oligosaccharide mixture significantly increased the concentration of acetic acid, propionic acid, butyric acid and total SCFAs in the fermentation broth [52], which was in line with the observation of the elevated SCFA production of supernatant contents caused by POS_H1_ in the current study. It has been reported that acetate and propionate were mainly metabolized by *Bacteroidetes*, while butyrate was mainly metabolized by *Firmicutes* [62,63]. SCFAs are produced by fermenting dietary fibers through gut microbes, so their changes are often associated with changes in the gut microbiota. High-throughput sequencing results showed that POS_H1_ increased the abundance of *Firmicutes* and *Bacteroidetes* in the intestinal flora, which may be the main reason for the increase in SCFAs in P24. It has been reported that butyric acid alleviated abnormal cholesterol metabolism in HF-diet ApoE knockout mice by upregulating the gene expression of macrophage ABCA1 [18]. The concentration of butyric acid in the P24 group was significantly higher than that in the N24 group, which may have been the reason for significantly promoting effects of P24on cholesterol efflux in our previous findings [28].

## 5. Conclusions

POS_H1_ altered the composition of the gut microbiota and its fermentation products. POS_H1_ significantly changed the composition of fecal gut microbiota at the phylum, genus and species levels; promoted the growth of bacteria related to cholesterol metabolism (*Bacteroidetes*, *Bifidobacterium* and *Lactobacillus*); and inhibited the growth of potential pathogenic bacteria such as *Escherichia coli* and *Enterococcus*, showing good probiotic activity.

POS_H1_ altered the composition of gut microbiota fermentation products, causing significant changes in 221 non-target fermentation products. Through KEGG pathway enrichment and fermentation product screening, the abundance of three fermentation products related to cholesterol metabolism (adenosine monophosphate, cyclic adenosine monophosphate and guanosine) in the P24 group was significantly higher than that in the N24 group.

POS_H1_ significantly promoted the secretion of SCFAs in the fermentation products of intestinal flora and increased the concentrations of acetic acid, propionic acid, isobutyric acid, butyric acid, isovaleric acid and valeric acid.

This is the first study to reveal the regulatory mechanisms of chemically prepared citrus POS on cholesterol metabolism via an integrative analysis of the gut microbiota. Our results demonstrated that POSs might be promising substrates for cholesterol metabolism regulation; however, further clinical research is recommended to elucidate the detailed regulatory pathways of POS on cholesterol metabolism.

## Figures and Tables

**Figure 1 nutrients-16-02002-f001:**
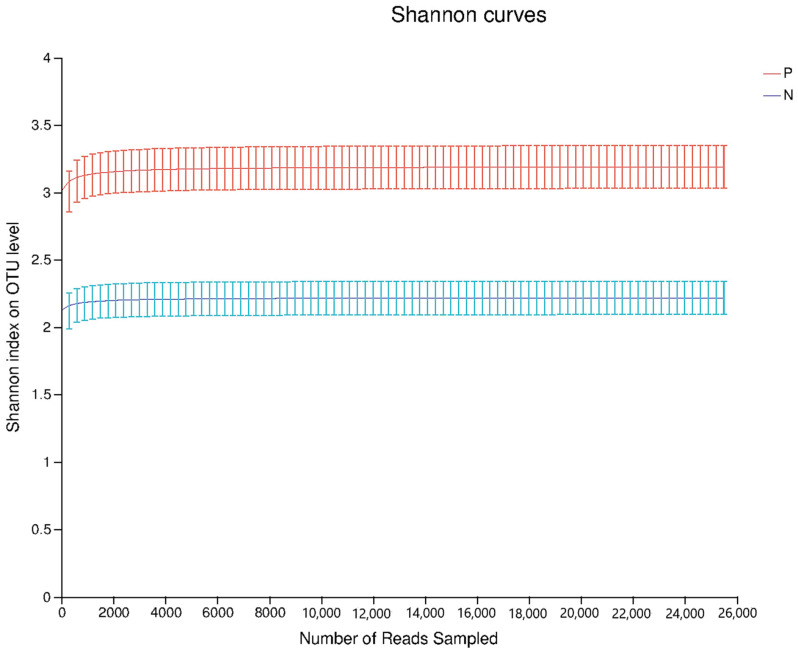
Comparison of POS_H1_ and negative control rarefaction curves.

**Figure 2 nutrients-16-02002-f002:**
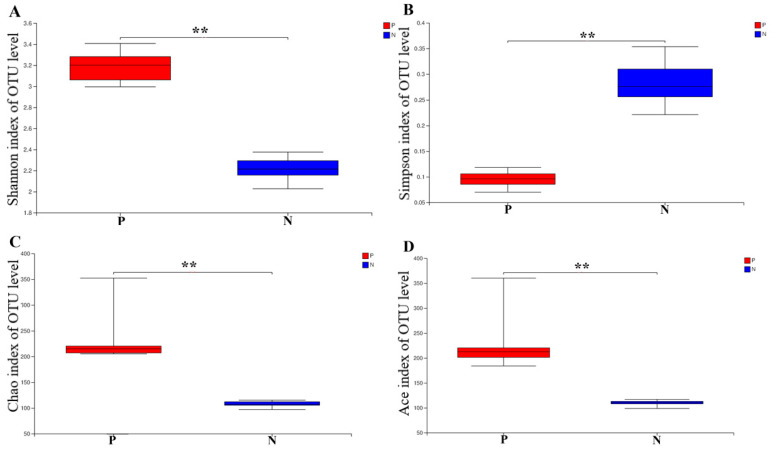
The α-diversity of gut microbiota after the POS_H1_ fermentation: (**A**) Shannon index, (**B**) Simpson index, (**C**) Chao index and (**D**) Ace index. Values are presented as the means ± SEM. ** *p* < 0.01.

**Figure 3 nutrients-16-02002-f003:**
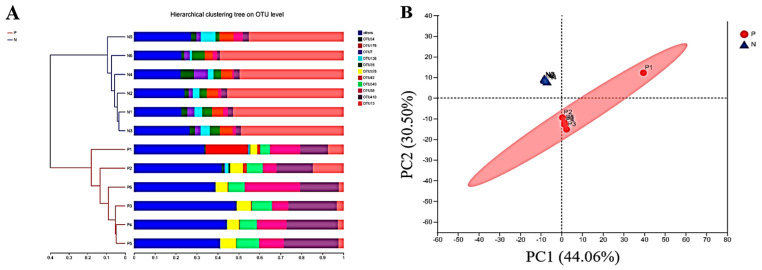
Hierarchical cluster (**A**) and principal component (**B**) analysis of gut microbiota after the POS_H1_ fermentation. Values are presented as the means ± SEM.

**Figure 4 nutrients-16-02002-f004:**
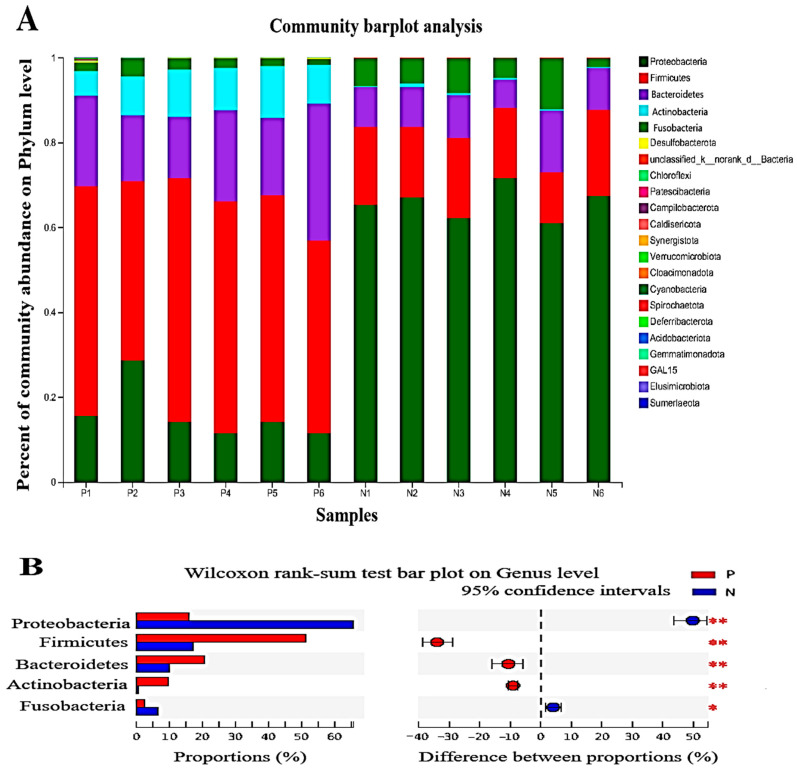
Percent of community abundance at the phylum level (**A**) and microorganism with statistical differences between groups on phylum level (**B**) after the POS_H1_ fermentation. Values are presented as the means ± SEM. * *p* < 0.05 and ** *p* < 0.01.

**Figure 5 nutrients-16-02002-f005:**
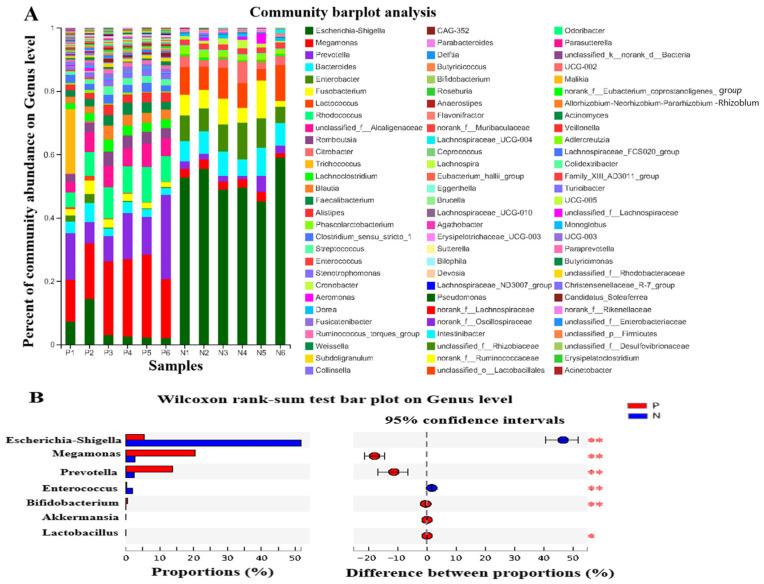
Percent of community abundance on genus level (**A**) and microorganism with statistical differences between groups on genus level (**B**) after the POS_H1_ fermentation. Values are presented as the means ± SEM. * *p* < 0.05 and ** *p* < 0.01.

**Figure 6 nutrients-16-02002-f006:**
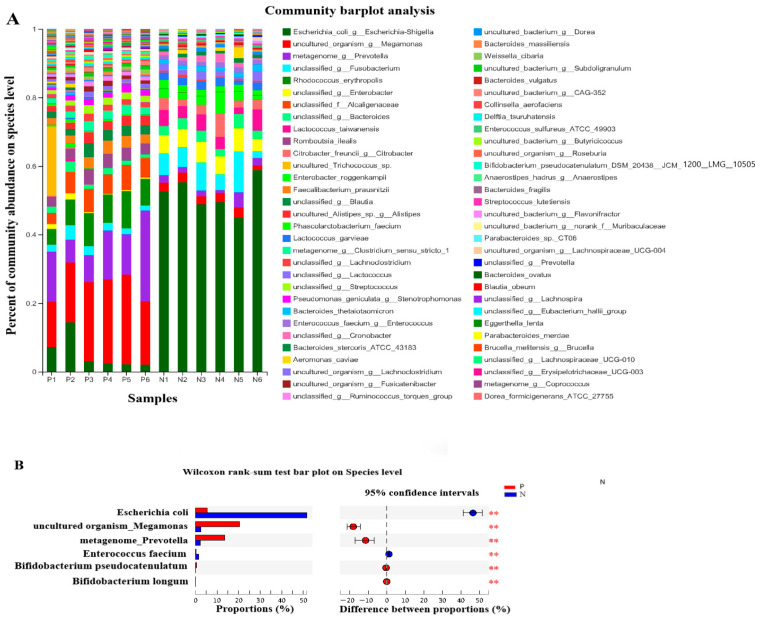
Percent of community abundance on species level (**A**) and microorganism with statistical differences between groups on species level (**B**) after the POS_H1_ fermentation. Values are presented as the means ± SEM. ** *p* < 0.01.

**Figure 7 nutrients-16-02002-f007:**
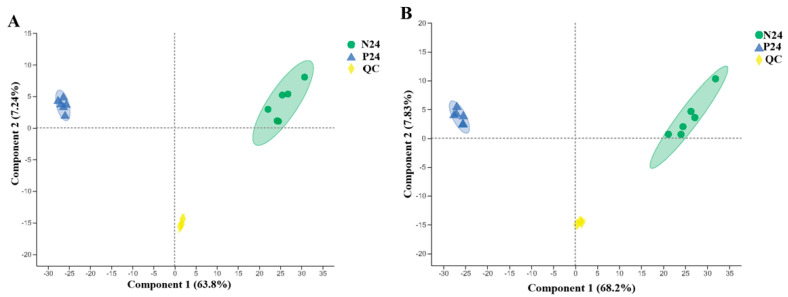
PLS-DA score chart of supernatants in P24 and N24 groups. (**A**) Positive ion-mode score chart (R^2^ = 0.996, Q^2^ = 0.979). (**B**) Negative ion-mode score chart (R^2^ = 0.995, Q^2^ = 0.986).

**Figure 8 nutrients-16-02002-f008:**
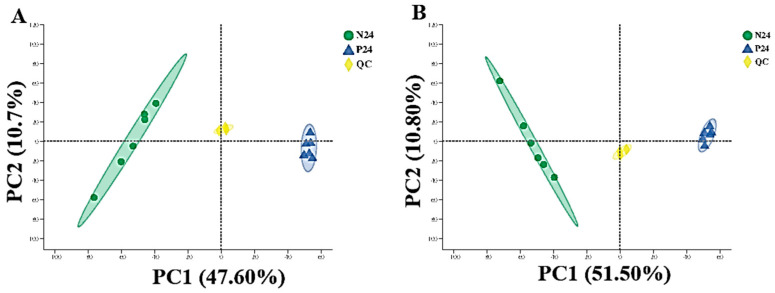
Principal component analysis of supernatants in P24 and N24 groups. (**A**) Positive ion-mode score chart. (**B**) Negative ion-mode score chart. Values are presented as the means ± SEM.

**Figure 9 nutrients-16-02002-f009:**
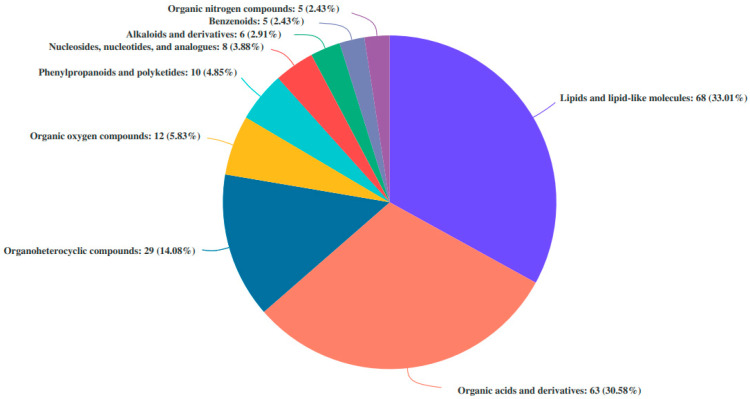
Human Metabolome Database (HMDB) compound classification of metabolites with significant differences between P24 and N24 groups at the superclass level.

**Figure 10 nutrients-16-02002-f010:**
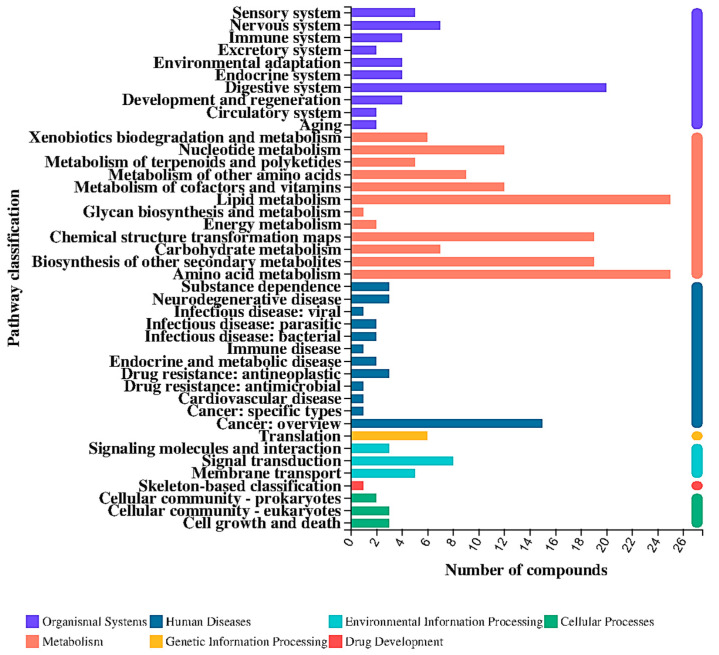
KEGG pathway classification of metabolites with significant differences between P24 and N24 groups at the superclass level.

**Figure 11 nutrients-16-02002-f011:**
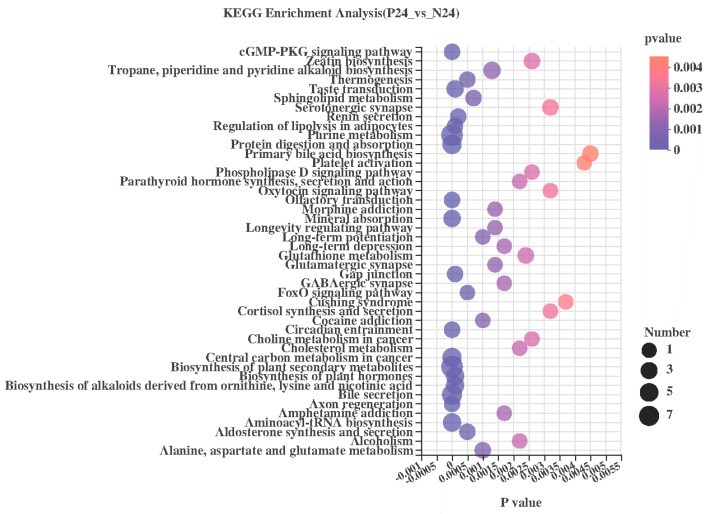
KEGG pathway enrichment analysis for metabolites with significant differences between P24 and N24 groups. Values are presented as the means ± SEM. *p* < 0.05.

**Figure 12 nutrients-16-02002-f012:**
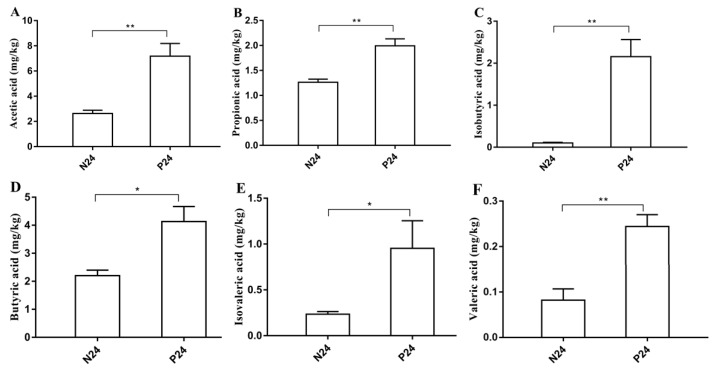
Comparison of SCFAs profiles between P24 and N24 groups. The SCFAs (acetic acid (**A**), propionic acid (**B**), isobutyric acid (**C**), butyric acid (**D**), isovaleric acid (**E**) and valeric acid (**F**)) levels among two groups were detected and are reported as mg per 1 kg of the contents. Values are presented as the means ± SEM. * *p* < 0.05 and ** *p* < 0.01.

**Table 1 nutrients-16-02002-t001:** Metabolites with significant differences between P24 and N24 groups.

Metabolite	KEGG Pathway Description	Fold Change (P24/N24)	*p*-Value	VIP Value
Glycocholic Acid	Bile secretion; cholesterol metabolism; secondary bile acid biosynthesis; primary bile acid biosynthesis	1.17	0.001	1.24
Taurocholic acid	Cholesterol metabolism; taurine and hypotaurine metabolism; secondary bile acid biosynthesis; bile secretion; primary bile acid biosynthesis	2.20	0.000	2.34
Sphinganine	Sphingolipid metabolism; metabolic pathways; sphingolipid signaling pathway	0.61	0.000	2.21
CGMP	Purine metabolism; bile secretion; salivary secretion; gap junction; thermogenesis; vascular smooth muscle contraction; oxytocin signaling pathway; circadian entrainment; regulation of lipolysis in adipocytes; aldosterone synthesis and secretion; renin secretion; cGMP-PKG	1.68	0.000	2.16
	Signaling pathway; olfactory transduction; long-term depression; phototransduction; platelet activation			
Adenine	Purine metabolism; zeatin biosynthesis	2.64	0.000	2.57
Inosine	ABC transporters; purine metabolism	1.17	0.000	1.14
Sphingosine	Sphingolipid metabolism; metabolic pathways; sphingolipid signaling pathway; apoptosis; necroptosis	0.82	0.006	1.01
3-ketosphinganine	Sphingolipid metabolism	0.24	0.000	2.57
3a,7a,12a-Trihydroxy-5b-cholestan-26-al	Primary bile acid biosynthesis	0.83	0.000	1.12
L-Phenylalanine	Mineral absorption; biosynthesis of secondary metabolites; aminoacyl-tRNA biosynthesis; cyanoamino acid metabolism; tropane, piperidine and pyridine alkaloid biosynthesis; ABC transporters; biosynthesis of various secondary metabolites—part 2; biosynthesis of amino acids; glucosinolate biosynthesis; 2-oxocarboxylic acid metabolism; phenylpropanoid biosynthesis; biosynthesis of plant hormones; phenylalanine metabolism; protein digestion and absorption; phenylalanine, tyrosine and tryptophan biosynthesis; biosynthesis of alkaloids derived from ornithine, lysine and nicotinic acid; biosynthesis of alkaloids derived from shikimate pathway; central carbon metabolism in cancer; biosynthesis of phenylpropanoids; biosynthesis of plant secondary metabolites	0.43	0.000	1.86
Guanine	Purine metabolism	398.08	0.000	2.90
Guanosine	ABC transporters; purine metabolism	2.73	0.000	2.70
L-5-Hydroxytryptophan	Serotonergic synapse; biosynthesis of alkaloids derived from shikimate pathway; tryptophan metabolism; axon regeneration	1.30	0.000	1.48
5′-Deoxy-5′-(methylthio) adenosine	Cysteine and methionine metabolism; biosynthesis of plant secondary metabolites; zeatin biosynthesis	1.23	0.000	1.32
L-Carnitine	Bile secretion; thermogenesis	1.25	0.000	1.25
Glycerophosp	Ether lipid metabolism; glycerophospholipid	2.40	0.000	2.73
hocholine	metabolism; choline metabolism in cancer			
Spermine	Bile secretion; glutathione metabolism; arginine and proline metabolism; beta-alanine metabolism	0.25	0.000	2.26
Cadaverine	Lysine degradation; biosynthesis of secondary metabolites; microbial metabolism in diverse environments; tropane, piperidine and pyridine alkaloid biosynthesis; glutathione metabolism; protein digestion and absorption; biosynthesis of alkaloids derived from ornithine, lysine and nicotinic acid; biosynthesis of plant secondary metabolites	0.03	0.000	2.97
2-Hydroxycinnamic acid	Biosynthesis of secondary metabolites; metabolic pathways; phenylpropanoid biosynthesis; microbial metabolism in diverse environments; phenylalanine metabolism	1.13	0.029	1.02
Adenosine monophosphate	Biosynthesis of secondary metabolites; purine metabolism; longevity regulating pathway; zeatin biosynthesis; Parkinson’s disease; regulation of lipolysis in adipocytes; aldosterone synthesis and secretion; renin secretion; cortisol synthesis and secretion; parathyroid hormone synthesis, secretion and action; biosynthesis of alkaloids derived from histidine and purine; taste transduction; morphine addiction; biosynthesis of plant secondary metabolites; metabolic pathways; FoxO signaling pathway; cGMP-PKG signaling pathway; olfactory transduction; cAMP signaling pathway; biosynthesis of plant hormones; antifolate resistance; Cushing syndrome; mTOR signaling pathway; PI3K-Akt signaling pathway; AMPK signaling pathway	6.11	0.000	2.56
L-Fucose	Microbial metabolism in diverse environments; amino sugar and nucleotide	0.66	0.000	1.75
	sugar metabolism; quorum sensing; fructose and mannose metabolism; two-component system; C-type lectin receptor signaling pathway			
N2-Acetyl-L-ornithine	2-Oxocarboxylic acid metabolism; arginine biosynthesis; biosynthesis of secondary metabolites; biosynthesis of amino acids	1.15	0.000	1.03
cAMP	Human T-cell leukemia virus 1 infection; cell cycle—yeast; Chagas disease (American trypanosomiasis); meiosis—yeast; oocyte meiosis; purine metabolism; inflammatory mediator regulation of TRP channels; Rap1 signaling pathway; Ras signaling pathway; pancreatic secretion; gap junction; MAPK signaling pathway; insulin secretion; human papillomavirus infection; apelin signaling pathway; longevity regulating pathway—multiple species; phospholipase D-signaling pathway	1.17	0.000	1.17
N-Acetylornithine	2-Oxocarboxylic acid metabolism; arginine biosynthesis; biosynthesis of secondary metabolites; biosynthesis of amino acids	3.00	0.000	2.56
5-Hydroxy-L-tryptophan	Serotonergic synapse; biosynthesis of alkaloids derived from shikimate pathway; tryptophan metabolism; axon regeneration	1.24	0.000	1.33
LysoPC (16:1(9Z)/0:0)	Glycerophospholipid metabolism; choline metabolism in cancer	4.19	0.000	2.16
LysoPC (16:0)	Glycerophospholipid metabolism; choline metabolism in cancer	3.88	0.000	1.86
LysoPC (14:1(9Z))	Glycerophospholipid metabolism; choline metabolism in cancer	5.45	0.000	2.06
L-Methionine	Mineral absorption; biosynthesis of secondary metabolites; 2-oxocarboxylic acid metabolism; cysteine and methionine metabolism; biosynthesis of amino acids; glucosinolate biosynthesis; protein digestion and absorption; aminoacyl-tRNA biosynthesis; antifolate resistance; central carbon metabolism in cancer; biosynthesis of plant hormones; biosynthesis of plant secondary metabolites	0.06	0.000	2.07
P-salicylic acid	Degradation of aromatic compounds;	1.27	0.000	1.20
	biosynthesis of secondary metabolites; microbial metabolism in diverse environments; aminobenzoate degradation; ubiquinone and other terpenoid-quinone biosynthesis; toluene degradation; bisphenol degradation; benzoate degradation; folate biosynthesis; benzoic acid family; biosynthesis of phenylpropanoids			
Pseudoegonine	Tropane, piperidine and pyridine alkaloid biosynthesis;	0.27	0.000	1.96
L-Tryptophan	biosynthesis of secondary metabolites; glycine, serine and threonine metabolism; biosynthesis of various secondary metabolites—part 2; African trypanosomiasis; indole alkaloid biosynthesis; glucosinolate biosynthesis; Serotonergic synapse; biosynthesis of alkaloids derived from shikimate pathway; biosynthesis of phenylpropanoids; biosynthesis of plant secondary metabolites; metabolic pathways; 2-oxocarboxylic acid metabolism; biosynthesis of amino acids; axon	3.66	0.000	2.64
Citric acid	Biosynthesis of various secondary metabolites—part 3; carbon metabolism; biosynthesis of secondary metabolites; citrate cycle (TCA cycle); glyoxylate and dicarboxylate metabolism; glucagon signaling pathway; biosynthesis of alkaloids derived from terpenoid and polyketide; biosynthesis of alkaloids derived from histidine and purine; taste transduction; biosynthesis of alkaloids derived from shikimate pathway; biosynthesis of terpenoids and steroids; biosynthesis of phenylpropanoids; biosynthesis of plant secondary metabolites; metabolic pathways; microbial metabolism in diverse environments; 2-oxocarboxylic acid metabolism; biosynthesis of amino acids; biosynthesis of alkaloids derived from	1.48	0.000	2.02
	ornithine, lysine and nicotinic acid; carbon fixation pathways in prokaryotes; two-component system; biosynthesis of siderophore group nonribosomal peptides; alanine, aspartate and glutamate metabolism; central carbon metabolism in cancer; biosynthesis of plant hormones			
Uric acid	bile secretion; purine metabolism; microbial metabolism in diverse environments	1.61	0.000	1.71
L-asparagine	Mineral absorption; biosynthesis of secondary metabolites; cyanoamino acid metabolism; biosynthesis of amino acids; alanine, aspartate and glutamate metabolism; protein digestion and absorption; aminoacyl-tRNA biosynthesis; central carbon metabolism in cancer; biosynthesis of plant secondary metabolites	7.33	0.000	2.55
L-glutamate	Biosynthesis of various secondary metabolites—part 3; carbon metabolism; arginine and proline metabolism; glutathione metabolism; carbapenem biosynthesis; taste transduction; alanine, aspartate and glutamate metabolism; nicotine addiction; protein digestion and absorption; phospholipase D-signaling pathway; ferroptosis; glyoxylate and dicarboxylate metabolism; taurine and hypotaurine metabolism; proximal tubule bicarbonate reclamation; D-glutamine and D-glutamate metabolism; ABC transporters; porphyrin and chlorophyll metabolism; neuroactive ligand-receptor interaction; Huntington disease; spinocerebellar ataxia; amyotrophic lateral sclerosis (ALS); GABAergic synapse; biosynthesis of alkaloids derived from ornithine, lysine and nicotinic acid; retrograde endocannabinoid signaling; amphetamine addiction; synaptic vesicle cycle; biosynthesis of plant secondary metabolites; metabolic pathways; arginine biosynthesis; gap junction	1.29	0.000	1.48
	Microbial metabolism in diverse environments; 2-oxocarboxylic acid metabolism; histidine metabolism; neomycin, kanamycin and gentamicin biosynthesis; biosynthesis of amino acids; FoxO signaling pathway; alcoholism; biosynthesis of secondary metabolites; C5-branched dibasic acid metabolism; glutamatergic synapse; cocaine addiction; butanoate metabolism; nitrogen metabolism; two-component system; circadian entrainment; Aminoacyl-tRNA biosynthesis; central carbon metabolism in cancer; Long-term potentiation; Long-term depression			
LysoPC(18:0)	Ether lipid metabolism; glycerophospholipid metabolism; choline metabolism in cancer; metabolic pathways	1.61	0.010	1.16
Phenylacetic acid	Tropane, piperidine and pyridine alkaloid biosynthesis; phenylalanine metabolism; biosynthesis of alkaloids derived from ornithine, lysine and nicotinic acid	0.72	0.000	1.89

## Data Availability

Data are included in the article and Appendix A; further inquiries can be directed to the corresponding author.

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
