# Peer review of "Regulations of Citrus Pectin Oligosaccharide on Cholesterol Metabolism: Insights from Integrative Analysis of Gut Microbiota and Metabolites"

_nutrients, 2024, doi:10.3390/nu16132002_

Round 1

Reviewer 1 Report

Comments and Suggestions for Authors

Haijuan et al presented a study entitled Regulations of Citrus Pectin oligosaccharide on Cholesterol Metabolism: Insights from Integrative Analysis of Gut Microbiota and Metabolites.

The title seems intresting and is worthy for publication, but the authors haven’t presented this study in a professional way.

The authors must revise the manuscript according to the following comments.

Abstract:

Our previous studies demonstrated that the cholesterol-lowering effects of citrus pectin oligosaccharides (POS) from a chemical degradation method were correlated to intestinal flora.

This sentence doesn’t make any sense and should be removed from the abstract.

Write clear aim of your study, what was the aim of this study.

The abstract should be written as follow.

Background, Aim, Methods, Results and conclusion.

Introduction

Line 37-48.

The whole paragraph should be revised .

The references from 1 to 10 are from year

1.     1984.

2.     2020.

3.     2017

4.     2017.

5.     2010.

6.     2012.

7.     2016.

8.     2021.

9.     2019.

10.   2018.

There is no single reference of 2022,2023 and even 2024 as there are thousands of latest articles present on this topic specifically Gut Microbiome which is a very hot topic now adays.

The authors must read and cite the following articles which are published in 2023 and 2024 and the introduction would be improved after reading and citing them.

1)     Elucidating the Role of Diet in maintaining the Gut Health to Reduce the Risk of Obesity, Cardiovascular and other age-related Inflammatory Diseases: Recent Challenges and Future Recommendations.

2)     Dietary Implications of the Bidirectional Relationship between the Gut Microflora and Inflammatory Diseases with special emphasis on Irritable Bowel Disease: Current and Future Perspective.

3) Genome investigation and Functional Annotation of Lactiplantibacillus plantarum YW11 Revealing Streptin and Ruminococcin-A as Potent Nutritive Bacteriocins against Gut Symbiotic Pathogens.

Line 49-52

The essential role of gut microbiota and their metabolites in cholesterol metabolism  is receiving attention. For gut microflora, Bacteroidetes, as a symbiotic gut bacteria, have 50 been proven to possess cholesterol-lowering properties [11]. Moreover, the main probiot ics lactobacilli and bifidobacteria are reported to be effective in reducing cholesterol levels [12, 13].

References are of 2006, 2007 and 2013 while bacterial metabolites have been extensively studied via in vivo, in vitro and even computationally recently which provides significant information that how it can regulate the Gut Microbiota.

The following articles should be read and cited for improvement of these paragraphs.

1)  Dose-dependent Production of Linoleic Acid Analogues in food derived Lactobacillus plantarum K25 and in silico Characterization of Relevant Reactions.

2) Conversion of linoleic acid to various fatty acid metabolites by Lactobacillus plantarum 13-3 and in silico characterization of the prominent reactions.

3)  In silico characterization of linoleic acid biotransformation to rumenic acid in food derived Lactobacillus plantarum YW11.

Third paragraph in Introduction section,

Paragraph no 19 is from 1995, The authors should cite recent references of the last 5 years.

Methodology

Well explained

Results

Well elaborated

Discussion

Recent literature should be cited

Figures quality is good,

Conclusions

Remove references from Conclusion

Write 2-3 lines of future perspective of your study.

Author Response

Reviewer: 1

Comments:

Haijuan et al presented a study entitled Regulations of Citrus Pectin oligosaccharide on Cholesterol Metabolism: Insights from Integrative Analysis of Gut Microbiota and Metabolites.

The title seems interesting and is worthy for publication, but the authors haven’t presented this study in a professional way.

The authors must revise the manuscript according to the following comments.

  1. Abstract:

Our previous studies demonstrated that the cholesterol-lowering effects of citrus pectin oligosaccharides (POS) from a chemical degradation method were correlated to intestinal flora.

This sentence doesn’t make any sense and should be removed from the abstract.

Write clear aim of your study, what was the aim of this study.

The abstract should be written as follow.

Background, Aim, Methods, Results and conclusion.

Response: We thank reviewer 1 for this suggestion. We’ve removed “Our previous studies demonstrated that the cholesterol-lowering effects of citrus pectin oligosaccharides (POS) from a chemical degradation method were correlated to intestinal flora.” from the abstract, please see Line 11.

The aim of this study was to reveal regulatory mechanisms of POS on cholesterol metabolism via integrative analysis of the gut microbiota. In order to explain the aim clearly, we’ve rewritten the aim in our revised manuscript, please see Line 14-17.

The abstract has been rewritten according to Reviewer 1’s comment. The abstract is written as follow: Background, Aim, Methods, Results and conclusion. Please see Line 11-29 in the attached revised manuscript.

  1. Introduction
    • Line 37-48.

The whole paragraph should be revised .

The references from 1 to 10 are from year

  1. 1984.
  2. 2020.
  3. 2017
  4. 2017.
  5. 2010.
  6. 2012.
  7. 2016.
  8. 2021.
  9. 2019.
  10. 2018.

There is no single reference of 2022,2023 and even 2024 as there are thousands of latest articles present on this topic specifically Gut Microbiome which is a very hot topic now adays.

The authors must read and cite the following articles which are published in 2023 and 2024 and the introduction would be improved after reading and citing them.

1) Elucidating the Role of Diet in maintaining the Gut Health to Reduce the Risk of Obesity, Cardiovascular and other age-related Inflammatory Diseases: Recent Challenges and Future Recommendations.

2) Dietary Implications of the Bidirectional Relationship between the Gut Microflora and Inflammatory Diseases with special emphasis on Irritable Bowel Disease: Current and Future Perspective.

3) Genome investigation and Functional Annotation of Lactiplantibacillus plantarum YW11 Revealing Streptin and Ruminococcin-A as Potent Nutritive Bacteriocins against Gut Symbiotic Pathogens.

Response: These recommended literature papers demonstrated an exhaustive literature survey about gut microbiome, health and disease. We have cited those publications in the introduction and No. 1, 2 and 4 in the reference section, please see Line 40-41.

2.2 Line 49-52

The essential role of gut microbiota and their metabolites in cholesterol metabolism is receiving attention. For gut microflora, Bacteroidetes, as a symbiotic gut bacteria, have been proven to possess cholesterol-lowering properties [11]. Moreover, the main probiotics lactobacilli and bifidobacteria are reported to be effective in reducing cholesterol levels [12, 13].

References are of 2006, 2007 and 2013 while bacterial metabolites have been extensively studied via in vivo, in vitro and even computationally recently which provides significant information that how it can regulate the Gut Microbiota.

The following articles should be read and cited for improvement of these paragraphs.

1) Dose-dependent Production of Linoleic Acid Analogues in food derived Lactobacillus plantarum K25 and in silico Characterization of Relevant Reactions.

2) Conversion of linoleic acid to various fatty acid metabolites by Lactobacillus plantarum 13-3 and in silico characterization of the prominent reactions.

3) In silico characterization of linoleic acid biotransformation to rumenic acid in food derived Lactobacillus plantarum YW11.

Response: These recommended literature papers provided sufficient supports about various metabolites could be produced in gut microbiota metabolism. We have cited those publications in the introduction and No. 14, 15 and 16 in the reference section, please see Line 56-57.

2.3 Third paragraph in Introduction section, Paragraph no 19 is from 1995, The authors should cite recent references of the last 5 years.

Response: We thank Reviewer 1 for this suggestion. We have deleted the reference from 1995, and cited a new publication from 2024 in the reference section, please see Line 69 and refence list No. 23 in Line 547.

  1. Methodology: well explained; Results: well elaborated; Discussion: recent literature should be cited.

Response: We thank Reviewer 1 for careful examination of our manuscript. We have deleted 6 old literatures and cited recent literatures instead. Updated references have been shown in our manuscript, please find reference list of No. 33 (a publication from 2024) in Line 563-564, No. 34 (a publication from 2022) in Line 565-566, No. 48 (a publication from 2024) in Line 591-592, No. 55 (a publication from 2023) in Line 605-606, No. 63 (a publication from 2024) in Line 621-622, and No. 64 (a publication from 2022) in Line 623-624.

  1. Figures quality is good; Conclusions: remove references from Conclusion, write 2-3 lines of future perspective of your study.

Response: We thank Reviewer 2 for this suggestion. References have been removed from Conclusion. “This is the first study to reveal regulatory mechanisms of chemically prepared citrus POS on cholesterol metabolism via integrative analysis of the gut microbiota. Our results demonstrated that POS might be promising substrates for cholesterol metabolism regulation, further clinical research is recommended to elucidate detailed regulatory pathways of POS on cholesterol metabolism.” has been added as a future perspective of our study in the revised manuscript, please see Line 487-491.

Reviewer 2 Report

Comments and Suggestions for Authors

The manuscript entitled “Regulations of Citrus Pectin Oligosaccharide on Cholesterol Metabolism: Insights from Integrative Analysis of Gut Microbiota and Metabolites” is an original study on the regulatory mechanisms of pectin on cholesterol metabolism through its action on the gut microbiota. The article is interesting and highlights experimental design. However, some points need to be corrected.

Comments:

Introduction:

Line 41: References 8–10 should also be inserted when mentioning “A large body of evidence has shown that changes in cholesterol levels are associated with alterations in the composition of the gut microbiota”.

Line 49: Similar to the previous point, in the sentence “The essential role of gut microbiota and their metabolites in cholesterol metabolism is receiving attention”, the authors must insert all the following studies that refer to this argument.

Line 55: The authors were not clear about the relationship between fatty acids, the intervention studied and the reduction in cholesterol levels. Please explain this connection.

Line 57: How is the expression of the ABCA1 gene, mentioned in the introduction, related to cholesterol metabolism or the intervention studied? This is not a gene studied in the article, therefore, its mention needs further clarification. The authors must also clarify this point in the manuscript.

Lines 61-74: In addition to the studies carried out by the authors themselves, are there other works that could contribute to the bibliographical survey of the introduction? If yes, please add them.

Methods:

Line 116: What is the justification for choosing multivariate analysis? What parameters were included in it? Although there is a brief explanation of this analysis in the supplementary material, the authors should include more information about it in the main manuscript.

Line 133: What is the α-diversity index and what is it for?

Results:

Line 172-182: What does the Shannon index measure? And the Simpson index, Chao Index, Ace index? Where were they explained in the methodology section? Please, if they have not already been mentioned previously, add them to the methodology.

Line 149: Where are the results (statistical description and graphical representation) regarding Pearson's correlation presented?

Discussion:

Line 336: Authors should begin by summarizing the main findings rather than inserting, in the first sentence, an idea from another study, without a prior original argument.

Authors must include the clinical relevance of the study. Why could this work contribute to the scientific community? Despite being a well-conducted study, the interpretation of the results is limited to the technical view of the methodology. The authors must go beyond the mechanistic view.

Author Response

Reviewer: 2

Comments:

The manuscript entitled “Regulations of Citrus Pectin Oligosaccharide on Cholesterol Metabolism: Insights from Integrative Analysis of Gut Microbiota and Metabolites” is an original study on the regulatory mechanisms of pectin on cholesterol metabolism through its action on the gut microbiota. The article is interesting and highlights experimental design. However, some points need to be corrected.

  1. Introduction:

Line 41: References 8–10 should also be inserted when mentioning “A large body of evidence has shown that changes in cholesterol levels are associated with alterations in the composition of the gut microbiota”.

Line 49: Similar to the previous point, in the sentence “The essential role of gut microbiota and their metabolites in cholesterol metabolism is receiving attention”, the authors must insert all the following studies that refer to this argument.

Response: We thank reviewer 2 for careful examination of our manuscript. We have inserted those references refer to “Strong correlations were observed between changes in cholesterol levels and microbiota composition [8-10]” and “A large body of evidence has shown that intestinal bacteria metabolites regulate the host metabolism positively [11-16]”, respectively, please see Lines 44-45 and Lines 52-53 in the revised manuscript.

Line 55: The authors were not clear about the relationship between fatty acids, the intervention studied and the reduction in cholesterol levels. Please explain this connection.

Response: The relationship between fatty acids, the intervention studied and the reduction in cholesterol levels is explained as follows:

According to our previous report, the cholesterol-lowering effects of citrus POS are modulated by specific bacterial groups together with their metabolites [1]. In our current study, the three main short chain fatty acids (SCFAs) of gut microbiota metabolites, acetic acid, propionic acid and butyric acid in metabolites were increased after POS fermentation. Thus, “As the three main short chain fatty acids (SCFAs) of gut microbiota metabolites, acetic acid, propionic acid and butyric acid were found to reduce plasma total cholesterol (TC) levels in hamster by promoting cholesterol decomposition and efflux” was cited here to provide potential regulatory mechanisms of POS on cholesterol mechanisms.

[1] Hu, H. J.; Zhang, S. S.; Liu, F. X.; Zhang, P. P.; Muhammad, Z.; Pan, S. Y., Role of the Gut Microbiota and Their Metabolites in Modulating the Cholesterol-Lowering Effects of Citrus Pectin Oligosaccharides in C57BL/6 Mice. Journal of Agricultural and Food Chemistry 2019, 67 (43), 11922-11930.

Line 57: How is the expression of the ABCA1 gene, mentioned in the introduction, related to cholesterol metabolism or the intervention studied? This is not a gene studied in the article. Therefore, its mention needs further clarification. The authors must also clarify this point in the manuscript.

Response: Sorry for not elucidating the point clearly. The full name of ABCA1 is adenosine triphosphate (ATP)-binding cassette transporter A1 (ABCA1). The ATP-binding cassette transporters ABCA1 is key mediators of macrophage cholesterol efflux[1]. Cholesterol metabolism could be promoted by upregulating the expression of the ABCA1 gene to enhance cholesterol efflux [1]. In order to express it clearly, the explanation was added in the revised manuscript, please see Line 62-65.

[1] Gelissen, I. C.; Harris, M.; Rye, K. A.; Quinn, C.; Brown, A. J.; Kockx, M.; Cartland, S.; Packianathan, M.; Kritharides, L.; Jessup, W. ABCA1 and ABCG1 synergize to mediate cholesterol export to apoA-I. Arterioscler. Thromb. Vasc. Biol. 2006, 26, 534−540.

Lines 61-74: In addition to the studies carried out by the authors themselves, are there other works that could contribute to the bibliographical survey of the introduction? If yes, please add them.

Response: There are 2 important bibliographies to our investigation, we’ve referred them according to Reviewer 2’s suggestion. Thus, “As potential prebiotics, pectin and chitosan-oligosaccharide have been proven to promote cholesterol metabolism by modifying intestinal flora compositions and SCFAs profiles [24, 25]” has been added in the introduction, please see Line 69-71.

  1. Methods:

Line 116: What is the justification for choosing multivariate analysis? What parameters were included in it? Although there is a brief explanation of this analysis in the supplementary material, the authors should include more information about it in the main manuscript.

Response: The metabolomics data obtained from nutritional intervention studies is too complex to dig important information. Multivariate analysis (MVA) is a useful approach to find meaning in metabolomics datasets involves parameters such as Partial least squares discriminate analysis (PLS-DA), principal component analysis (PCA), partial least squares projection to latent structures (PLS), and Orthogonal partial least squares discriminate analysis (OPLS-DA) [1,2].

In our study, PLS-DA and PCA were employed in the MVA. PCA using an unsupervised method was applied to obtain an overview of the metabolic data, general clustering, trends, or outliers were visualized. PLS-DA was used for statistical analysis to determine global metabolic changes between comparable groups. We’ve added above information as a further explanation in the revised main manuscript, please see Lines 124-128.

The results of PLS-DA showed that the R2 and Q2 of the positive and negative ion score maps were both close to 1, indicating the model was stable and reliable with good predictive ability. Under this model, the supernatant fermentation products of the P24 and N24 groups were well separated. Those contents were shown in our submitted manuscript, please see Lines 270-274.

PCA analysis indicated that the non-targeted metabolomics data of the P24 and N24 groups had a clear boundary, and the sample distribution between the groups was far away and the similarity was low. Those results indicated that the composition of the fermentation products of the two groups was significantly different (P < 0.05), showing that the addition of POSH1 to the medium caused metabolic changes in the overall gut microbiota. Those contents were shown in our submitted manuscript, please see Line 279-284.

In addition, there are 2 typing errors in the supplementary materials, “Orthogonal Partial least squares discriminate analysis (OPLS-DA)” has been corrected as “Partial least squares discriminate analysis (PLS-DA)”, and “R2 and Q2” has been corrected as “R2 and Q2” in the revised supplementary materials, please see Line 124 and Line 127.

[1] Worley, B. and R. Powers, Multivariate analysis in metabolomics. Current metabolomics, 2013. 1(1): p. 92-107.

[2] Westerhuis, J.A., et al., Multivariate paired data analysis: multilevel PLSDA versus OPLSDA. Metabolomics, 2010. 6: p. 119-128.

Line 133: What is the α-diversity index and what is it for?

Response: Alpha diversity (α-diversity) is defined as the mean diversity of species in different sites or habitats within a local scale[1]. The Shannon index, Simpson index, Chao index, and Ace index were used to describe the α-diversity index. α-diversity is used to define the microbial diversity after POS addition.

[1] Andermann, T., et al., Estimating alpha, beta, and gamma diversity through deep learning. Frontiers in plant science, 2022. 13: p. 839407

  1. Results:

Line 172-182: What does the Shannon index measure? And the Simpson index, Chao Index, Ace index? Where were they explained in the methodology section? Please, if they have not already been mentioned previously, add them to the methodology.

Response: We thank reviewer 2 for careful examination of our manuscript. The Shannon index, Simpson index, Chao index, and Ace index were used to describe the α-diversity index, which is employed to reveal the community richness after POS intervention. Sorry, the explanation was missed previously. We’ve added in the revised supplementary materials, please see Line 70-71.

Line 149: Where are the results (statistical description and graphical representation) regarding Pearson's correlation presented?

Response: We thank reviewer 2 for careful examination of our manuscript. Pearson's correlation was not used in our current study. Thus, “Pearson correlation analysis was used for correlation test” was deleted in our revised manuscript, please see Line 157.

  1. Discussion:

Line 336: Authors should begin by summarizing the main findings rather than inserting, in the first sentence, an idea from another study, without a prior original argument.

Response: We thank Reviewer 2 for this suggestion. We’ve added the summary of our main findings in the discussion in the revised manuscript, please see Lines 344-351.

  1. Authors must include the clinical relevance of the study. Why could this work contribute to the scientific community? Despite being a well-conducted study, the interpretation of the results is limited to the technical view of the methodology. The authors must go beyond the mechanistic view.

Response: We thank Reviewer 2 for this suggestion. The current research is a consecutive study from our previous in vitro and in vivo observations [1-3]. The main purpose of this study is to further elucidate the regulatory mechanisms of POS on cholesterol mechanisms via revealing changes of whole gut microflora composition and metabolites. This is the first study to reveal regulatory mechanisms of chemically prepared citrus POS on cholesterol metabolism via integrative analysis of the gut microbiota. Our results demonstrated that POS might be promising substrates for cholesterol metabolism regulation. For more clinical relevance, clinical research is needed. However, as a new substrate, POS is hard to get permission to do experiments with human so far. As a result, we couldn’t include more clinical relevance of the study. Clinical research using our POS samples is included in our research direction for further studies. We have included this statement in lines 487-491.

[1] Zhang, S.S., et al., Preparation and prebiotic potential of pectin oligosaccharides obtained from citrus peel pectin. Food Chemistry, 2018. 244: p. 232-237.

[2] Hu, H.J., et al., Role of the Gut Microbiota and Their Metabolites in Modulating the Cholesterol-Lowering Effects of Citrus Pectin Oligosaccharides in C57BL/6 Mice. Journal of Agricultural and Food Chemistry, 2019. 67(43): p. 11922-11930.

[3] Haijuan, H., Z. Shanshan, and P. Siyi, Characterization of citrus pectin oligosaccharides and their microbial metabolites as modulators of immunometabolism on macrophages. Journal of Agricultural and Food Chemistry, 2021. 69(30): p. 8403-8414.

Round 2

Reviewer 1 Report

Comments and Suggestions for Authors

The authors have revised the manuscript and the article can be accepted for publication.